# Plant Photocatalysts: Photoinduced Oxidation and Reduction Abilities of Plant Leaf Ashes under Solar Light

**DOI:** 10.3390/nano13152260

**Published:** 2023-08-06

**Authors:** Xiaoqian Ma, Jiao He, Yu Liu, Xiaoli Bai, Junyang Leng, Yi Zhao, Daomei Chen, Jiaqiang Wang

**Affiliations:** 1School of Chemical Sciences & Technology, Yunnan University, Kunming 650091, China; 18290855825@163.com (X.M.); hejiao@ynu.edu.cn (J.H.); m18213412529@163.com (X.B.); ljy262@163.com (J.L.); m18788591505@163.com (Y.Z.); 2School of Materials and Energy, Yunnan University, Kunming 650091, China; liuyu19990204@163.com

**Keywords:** plant leaf ash, photocatalysis, photodegradation, photoreduction, dehydrogenase-like

## Abstract

Plant leaf ashes were obtained via the high temperature calcination of the leaves of various plants, such as sugarcane, couchgrass, bracteata, garlic sprout, and the yellowish leek. Although the photosynthesis systems in plant leaves cannot exist after calcination, minerals in these ashes were found to exhibit photochemical activities. The samples showed solar light photocatalytic oxidation activities sufficient to degrade methylene blue dye. They were also shown to possess intrinsic dehydrogenase-like activities in reducing the colorless electron acceptor 2,3,5-triphenyltetrazolium chloride to a red formazan precipitate under solar light irradiation. The possible reasons behind these two unreported phenomena were also investigated. These ashes were characterized using a combination of physicochemical techniques. Moreover, our findings exemplify how the soluble and insoluble minerals in plant leaf ashes can be synergistically designed to yield next-generation photocatalysts. It may also lead to advances in artificial photosynthesis and photocatalytic dehydrogenase.

## 1. Introduction

Photosynthesis evolved very early, and is essential not only to all biological activities, but also to geological and climatic activities and, eventually, to human life [1]. Photosynthesis was also one of the main events in the early evolution of life on Earth, though it has long remained unresolved [2]. A natural leaf consists of elaborated structures and functional components, including light harvesting, photoinduced charge separation, and catalysis modules, which comprise the highly complex machinery for photosynthesis [3]. To look to natural photosynthesis for inspiration, the wisest way in which artificial photosynthesis can attempt to replicate the natural process of photosynthesis is by synthesizing leaflike hierarchical structures and analogous functional modules. This has been the subject of extensive research in the past decades, and significant progress has been achieved toward the design and construction of chemical assemblies that function as artificial reaction centers [4,5]. Though artificial photosynthesis is a promising way to store renewable energy in chemical bonds, it is still very challenging to integrate multiple chemical functions in a stable chemical architecture and, simultaneously, optimize the activity, stability, selectivity, and energetic efficiency of the catalysts [6]. Moreover, most research has only focused on the functional imitation of photosynthesis, and has neglected the structural effect. Actually, the entire structure of natural leaves strongly favors light harvesting [7]. Nevertheless, a variety of artificial photosynthetic systems containing electron donors and acceptors to mimic light-driven charge separation, which occurs in photosynthetic reaction centers, have been reported [8]. In particular, a high surface area and porosity, for maximizing active site exposure, are often incorporated into biological design elements [9].

On the other hand, in light of the increasingly serious energy and environmental problems, considerable efforts have been invested into developing photocatalysts capable of using the abundant solar light. However, the availability and use of these photocatalysts is limited, because of the high cost of their large-scale production, coupled with the tremendous difficulties involved with deliberate synthesis [10]. In contrast, green plants provide an ideal example of a process that has been optimized to harness solar energy efficiently; namely, photosynthesis. The energy-harvesting capabilities of plants can help point the way when it comes to the design of new and more efficient low-cost solar light photocatalysts. For example, chlorophyll derivatives, and related natural porphyrins, have been used in the photosensitization of TiO_2_ solar cells [11]. Photosynthetic cyanobacterial strains of the silica gels immobilized in mesoporous silica matrices exhibited a significant viability, even after three months [3,12,13]. In recent years, green leaves were applied as biotemplates to synthesize *morph*-TiO_2_, which exhibited activities for the degradation of rhodamine dye under UV and visible light irradiation [14]. Bacterial cellulose membranes [15], plant skins [16], leaves [17], natural rubber latex [18], hypocrellins [19], diatom [20], and cyanobacteria [21] have also been used as templates to fabricate mesoporous titania with good photocatalytic activity [22].

There is a large body of experimental evidence demonstrating heterogeneous photocatalysis using both synthetic and natural solids as photocatalysts [23]. For instance, the photofixation of dinitrogen using desert minerals has been reported [24]. Interestingly, it was assumed that photocatalytic reactions may take place on atmospheric aerosols, and the surfaces of natural solid dust, soil, and sand particles, under ambient conditions [25,26,27]. Minerals and metal ions have also been investigated as potential catalysts for prebiotic reactions [28,29,30]. Additionally, hydrated minerals were proposed as a ‘‘ready-made catalyst’’, attaching to proteins to form the first cofactors [31]. Recently, the observation of high-molecular-weight compounds from formamide attributed to nucleoside bases on TiO_2_ single-crystal surfaces, at an ultra-high vacuum and under soft UV, is direct proof of the mineral’s role in creating compounds of biological significance on earth and/or anywhere in the galaxy [32].

However, it is difficult not to be continually amazed that so little attention has been paid to the solar light photocatalytic activities of minerals in plant leaves after the removal of three key aspects (the antenna, reaction center, and quinine pool) of a primary process in natural photosynthesis. What differentiates our work from others is that we show that after the destruction of the photosynthesis systems in leaves, the calcined plant leaf ashes exhibited solar light photocatalytic activities. This is, to the best of the current author’s knowledge, the first report on the photocatalytic properties of the minerals in plant leaf ashes after the removal of three key aspects of a primary process in natural photosynthesis. The discovery of the photocatalytic activities of the minerals in plant leaf ashes may suggest that the minerals might have played important roles in catalyzing prebiotic reactions, and offer a new way to obtain photocatalysts, through planting instead of chemical synthesis [33].

On the other hand, the dehydrogenation of organic substrate is essential in most biomedical and chemical engineering uses. For instance, the dehydrogenase activity (DHA) provides important information on the biological activity in many systems. Thus, the reduction of 2,3,5-triphenyltetrazolium chloride (TTC) to triphenylformazan (TPF) has been commonly used to estimate the DHA in soil, and TTC has been widely used as a hydrogen receptor in the enzymatic activity test [34,35,36,37]. Artificial enzyme mimetics have been intensively studied, due to some considerable shortcomings, such as a relatively low stability, and the poor recovery of natural enzymes [30,38]. A large number of nanomaterials have been found to demonstrate intrinsic unexpected enzyme-like activity, including metal oxides [39], precious metals [40], carbon-based materials [41], etc. Many approaches have been taken to designing artificial, inorganic, and organic–inorganic composite structures that mimic the enzymatic function [42]. However, these nanostructures are chemically synthesized, such as FeS [43], dextran-coated CeO_2_ [44], Fe_3_O_4_ [45], carboxyl-modified graphene oxide [46], and AgM bimetallic alloy (M = Au, Pd, Pt) [47]. Attractively, two-dimensional SnSe has been reported to exhibit the efficient catalytic activity of some dehydrogenases in cell metabolism, and to exhibit the prominent features of natural enzymes, for the first time [48].

As light is a natural agent, it inarguably merits considerable attention as a unique environmentally compatible option for driving biocatalytic reactions. However, there are still limited studies on catalytically active enzymes requiring light for the actual reaction (photosystem, photolyase, decarboxylation, pchlide reductase) [49,50,51,52]. Differently to thermal activation, photochemical activations offer an attractive alternative to conventional strategies for complex biological and biomimetic processes. Accordingly, a photocatalytic artificial enzyme (photozyme) is an enzyme that catalyzes photochemical reactions [53]. It can catalyze chemical reactions under light irradiation, and has the advantages of high efficiency, controllability, and repeatability. For example, a robust synthetic oxidoreductase was successfully designed to mimic the catalytic performance of galactose oxidase, and related biocatalysts, under irradiation [54]. Moreover, a CdS–Pt nanoparticle system prepared via the peptide-mediated synthesis method, behaved as a robust inorganic mimic in the reduction of nitrate to nitrite, with an activity level that outperformed the nitrate reductase enzyme > 23-fold [55]. Au@NH_2_-MIL-125(Ti) exhibited oxidase mimic activity under sunlight conditions, and showed a better substrate affinity and reaction rate compared to natural enzymes [56]. The polyaniline nanofiber, a conductive polymer that mimics oxidase activity under photoactivation (sunlight), demonstratesd an efficient intrinsic light-activated oxidase-like activity. The CoS_2_/MoS_2_ nanosheet showed a simulated enzyme activity in the presence of sunlight, which not only degraded nearly 90% of the methylene blue, methyl orange, and rhodamine B within 1 h, but also killed some bacteria [57]. V_2_O_5_ nanorods have a catalytic activity similar to catecholoxidase, and exhibit excellent efficiency in the degradation of methylene blue under sunlight conditions [58].

The photocatalytic simulation of dehydrogenase activity has important scientific significance for further understanding the reaction mechanism of dehydrogenase, and improving its efficiency and stability, and also brings new enlightenment to the field of the photocatalytic simulation of enzymes. Herein, we show that plant leaf ashes obtained via simple calcination, without the addition of any external chemicals, and unaided by any chloroplasts or other cellular particles, had the ability to reduce TTC under solar light. In this featured photoreaction, three key aspects of a primary process in natural photosynthesis were absent, and the plant leaf ashes played the role of dehydrogenase. In contrast to all the previous reductions of TTC, which was not light-driven, the reduction of TTC over the plant leaf ashes was found to be efficient only under light irradiation. We obtained enzymatic mimics directly from plants in this work, rather than via a synthesis process. The dehydrogenase-like activities of the plant leaf ashes, once again, implied the probable fact that minerals might drive dehydrogenizing reactions in biological and chemical evolution.

## 2. Experimental

### 2.1. Preparation of Plant Leaf Ashes

The plant samples were firstly dried in air at 90 °C, and then the organics were removed via calcination at 500 °C for 6 to 8 h (to ensure complete calcination). The preparation method is quite special; it is a synthesis derived from the plant itself, rather than through the addition of chemicals.

### 2.2. Photocatalytic Activity Measurement

The photodegradation of methylene blue (MB) was carried out in a glass beaker as the photoreactor vessel. The reaction system, containing 50 mL of MB solution (10 ppm) and 25 mg ash, was magnetically stirred in the dark for 24 h, to reach the adsorption equilibrium of MB with the catalyst, and then exposed to solar light from 9:00 a.m. to 3:00 p.m. Where the solar intensity fluctuations were minimal on a sunny day, the solar radiation energy is 16.28–16.50 MJ·m^−2^. UV–vis absorption spectra were recorded (485 nm) at different intervals, using a spectrophotometer (Shimadzu UV 2401 PC), to monitor the reaction. The photodegradation yield of MB was defined as:(1)$$\mathrm{Photodegradation}\;\mathrm{yield}=\frac{C_{0}-C_{a}-C_{b}}{C_{0}}\;\times \;100$$

*C*_0_ is the initial MB concentration (10 ppm), *C_a_* is the concentration after photodegradation, and *C_b_* is the decrease in the concentration because of direct photolysis.

The photoreduction of TTC was carried out in a glass beaker, containing 50 mL of TTC solution (1 g/L) and 25 mg of the ash, under direct solar light, for 3 h. During this period, the conversion of TTC was analyzed via the detection of the absorption of TPF in aqueous acetone at the maximum absorptive wave of TPF (485 nm) using a UV–vis spectrophotometer (Shanghai Precision 722N, Shanghai, China). The photoreduction rate of TTC was defined as:(2)$$\mathrm{Photoreduction}\;\mathrm{rate}=\frac{C_{t}}{C}\;\times \;100$$

*C_t_* is the concentration of TPF after photoreduction, and *C* is the concentration of TPF when the TTC was completely hydrogenated.

### 2.3. Characterizations

TEM images of the samples were collected using a Phoenix JEOL JEM 2010F microscope, working at 200 kV, and equipped with an energy-dispersive X-ray analyzer. A Hitachi S-4300F scanning electron microscope (SEM) was used to investigate the morphology of the ashes. The Brunauer–Emmett–Teller (BET) specific surface area of all the samples was measured using the CO_2_ adsorption at 273.15 K, and the N_2_ adsorption at 77 K, with ASAP2020 (Micromeritics, Norcross, GA, USA). UV–vis diffuse reflectance spectra were measured at room temperature in air on a Shimadzu UV-2401PC photometer, ranging from 200 to 800 nm. X-ray powder diffraction (XRD) measurements were carried out using a Rigaku TTRIII instrument, with filtered Cu Kα radiation. Scans were performed in the 2θ range 5°–90°, with a scan rate of 10°/min. The element contents of the products were estimated using a Rigaku ZSX100e X-ray fluorescence spectrometer (XRF).

## 3. Results and Discussion

### 3.1. Preparation and Characterization of Different Plant Leaves

We collected 49 plant samples in Yunan Province. In addition, the Latin names of the plants are summarized in Table 1. Oxalates, silica, and carbonates were generally found to be the most major contributors to the plant leaf ashes [59]. Particularly, the diffuse reflectance UV–vis spectrum of ash sample from couchgrass (shown in Appendix A) clearly shows very broad absorption over 200 to 800 nm. As expected, the typical UV-vis spectrum of chlorophyll was not observed. This implies that the PS II was destroyed by calcinations. In the following, we selected garlic as an example for detailed investigation. XRD, BET-BJH, UV–Vis diffuse reflectance spectra, XRF, TEM, and SEM were also employed, to characterize these samples. The main element contents of the garlic sprout ashes were representatively determined via XRF measurement. Elements such as K, Cl, O, Ca, C, Mg, and so forth were detected (Table 2). KCl, SiO_2_, CaCO_3_, and so on were also observed via the XRD (Figure 1). The diffraction peaks of 28.3°, 40.5°, 50.2°, 66.4°, and 73.7° in Figure 1 correspond with the (200), (220), (222), (420), and (422) crystal planes of KCl (JCPDS 41-1476), respectively. The characteristic diffraction peaks at 20.9°and 28.3° belong to the (110) and (022) crystal planes of SiO_2_ (JCPDS 10-0348). The remaining diffraction peaks of 31.8°, 33.86°, and 37° correspond to the (220), (221) and, (302) crystal planes of CaCO_3_ (JCPDS 17-0763), respectively, without the presence of impurity peaks.

XPS measurement of the garlic sprouts (stem) also show the same elements with XRF regardless of different content (shown in Appendix A). 

As shown in Figure 2, the SEM images of the other ash samples also show interesting morphologies, such as tubular, plush-like, and hollow tubes, with rough surfaces. This suggests that it is worthwhile to conduct further study on the morphologies of the plant leaf ashes.

There are a wide variety of compounds in the plant leaf ashes; some of them could be potential efficiency photocatalysts. The existence of different ions may also promote photocatalytic activity, as mentioned in the literature [60,61]. Very interestingly, after the washing of the ashes, the decrease in the photodegradation yield proved the promotion roles of soluble ions in ashes (Figure 3). Therefore, these findings suggest that we should also pay more attention to the synergistical effects of the soluble and insoluble components in the design of photocatalysts in the future.

The presence of metal oxides in plant leaf ashes may also be beneficial. For instance, we found that adding a small amount of GeO_2_ powders into the reaction solution significantly enhanced titania-based photocatalytic activity in the degradation of dyes [62]. Thus, a synergistic strong interaction between various elements, particularly diverse ions and residual carbon species, in these plant leaf ashes would be beneficial to photocatalytic degradation. The differences in the catalytic activities of ashes from different parts of the tissue of one plant may also be due to the discrepancy in the absorption and enrichment of mineral elements from the soil among them. However, the chemical compositions of different types of plant leaf ashes are widely varied, which could lead to differences in their reaction behaviors.

Moreover, the diverse morphology (tubular, spherical, columnar, etc.) of these plant photocatalysts would provide a suitable reaction environment [63,64]. As the chemical compositions of plant ashes vary, because of different species, habitats, and growth conditions, their morphologies are also diverse. As an example, TEM and SEM images of sugarcane tops (*Saccharum officinarum*) ash are shown in Figure 4a,b, respectively. The TEM image reveals that this sample has a microtubule morphology, while the SEM suggests that it is composed of spherical particles, with diameters of 0.1–0.3 µm. The BJH pore size distribution (inset in Appendix A) shows primary pore size distributions, which implies irregular mesoporous channels in the ash samples. 

### 3.2. Photocatalytic Oxidation and Reduction Activities

As methylene blue is resistant to biodegradation and photodegradation under visible irradiation [65], the photoactivity of the plant leaf ashes were evaluated via the degradation of MB under solar light. It is seen that most of the ashes, including fungus ash, exhibited meaningful activity (Table 3). In particular, yellowish leek (*Allium tuberosum Rottler*) ashes exhibited the negligible activity of only 4.4%, but the high photocatalytic activity of bracteata (*Hyparrhenia bracteata*), oyster mushroom (*Pleurotus ostreatus*), couchgrass and sugarcane top ashes is impressive. As an example, the variation in the UV–Vis spectra of MB during photodegradation over sugarcane top ash under solar light is shown in Figure 5. It is seen that the absorption peaks corresponding to MB diminished under solar light, which indicates the degradation of MB. No new absorption bands, in particular the absorption bands of aromatic moieties and other similar intermediates, were observed. The time course of the photodegradation of MB over sugarcane top ash is shown in the insert in Figure 5. These ashes exhibited different photodegradation yields in the range of 4.4% to 82.6%.

In addition to the photocatalytic degradation of MB, the photocatalytic reduction of TTC to TPF was also found efficiently over all 49 types of plant ashes (Table 3) under solar light irradiation, while they did not exhibit any activity on TTC without solar light irradiation. So, this reduction process can be described as a photocatalytic reduction. It is seen that most of the plant leaf ashes, including fungus ashes, showed attractive activity, even though the rice hull (*Oryza satival*), danhuidajiezhu (*Inda sasa Singulispicula*), yeast, and bitter bamboo leaf ashes exhibited negligible activity. In particular, the photoreduction rates of TTC over bracteata, oyster mushroom, all the grass of tiger grass, ramalina canduplicans (*Fructificatio Schizoplylli Communis*), corn (*Zea mays Linn*) leaf, and garlic sprout (leaf) ashes are over 60%. The photoreduction rates of TTC in more than three-quarters of the ashes are in the range 20~60%. As an example, the photoreduction rate of TTC over bracteata ash under solar light is shown in Figure 6. After high temperature calcination, there should be no living cells or biological species left. By sharp contrast, chemically synthesized photocatalysts, such as Co-MCM-41, Co-SBA-15, Ce-FSM-16 [18], and commercial TiO_2_ (Degussa P25) could not convert TTC to TPF, even under solar light irradiation.

TTC dissolves in water as a colorless solution but, after reduction, it forms the red and insoluble tribenzyl (TPF). TPF is stable, and will not be automatically oxidized by oxygen in the air, so TTC is widely used as a hydrogen receptor for enzyme tests. The plant raw material photocatalysts synthesized in this study all showed photocatalytic TTC reduction activity, indicating, to some extent, that they had the effect of simulating dehydrogenase under light.

The negligible activity of the ashes from yellowish leaf may be caused by the cultivation style of this vegetable. It was cultivated under complete darkness, when its corresponding Chinese leek germinated. The yellow leaves imply the lack of chlorophyll, because no or less chlorophyll was synthesized. Interestingly, the ashes from different parts of plants exhibited different activities. Root and stem ashes generally exhibited much lower photodegradation yields than their corresponding leaf ashes. For example, the photodegradation yield of garlic sprout (leaf) ash (Table 3, Entry 7) is about three times that of garlic sprout (stem) (Table 3, Entry 8), and five times of garlic sprout (root) (Table 3, Entry 9). Therefore, the above results lead to a hypothesis that the plant leaf ashes of plant tissues with higher contents of chlorophyll would exhibit higher photocatalytic yields, and a higher chlorophyll content is more conducive to a high light-energy-capturing efficiency. Although no clear relationship can be observed between the amount of chlorophyll or the plant species and the photocatalytic activity of plant leaf ashes, the diverse plants would provide us with chances to obtain a highly efficient solar light photocatalyst in large-scale production.

On the other hand, summary of textural properties of Chinese leek (leaf and stem) and garlic sprouts (leaf and stem) ashes was listed in Appendix A. It was found that the specific surface area is not the most important factor affecting the performance of these ash samples (See Appendix A). For example, the BET surface area and pore volume of the couchgrass (*Elymus repens*) ash was 41.9 m^2^·g^−1^ and 14.7 cm^3^·g^−1^; the photodegradation rate of MB and the photoreduction of TTC is 80% and 25.8%, respectively. In contrast, the photodegradation rate of MB and the photoreduction of TTC is 40.7% and 67.9%, respectively, in spite of the very low BET surface area (5.8 m^2^·g^−1^). This implies that the plant leaf ashes did behave differently from conventional single or chemically synthesized photocatalysts.

A photozyme, a new type of nanozyme working under light irradiation, is an enzyme that catalyzes photochemical reactions [53,66]. Quite a few TiO_2_-based photocatalytic artificial enzymes were synthesized, and exhibit excellent peroxide-like specificity under visible light irradiation. Additionally, TiO_2_ could be used to simulate photolyase, and repair DNA damage [67]. Carbon nanodots containing histidine was also found to behave as an oxidation simulation nanozyme, to inhibit the growth of pathogenic Escherichia coli, and the inhibition rate exceeded 80% in the presence of visible light irradiation [68]. The plant leaf ashes are obtained via simple calcination, without the addition of any external chemicals, and had the ability to reduce TTC under solar light. Compared with two-dimensional SnSe [69], the first artificial dehydrogenases in cell metabolism, our plant leaf ashes exhibited dehydrogenase mimic activity only under solar light irradiation. Hence, we proposed the concept of simulated photocatalytic dehydrogenase.

Very interestingly, the photodegradation yields of 17 types of plant ash in Table 1 for the degradation of MB are linearly correlated with the photocatalytic reduction of TTC (Figure 7). This suggests that those plant ashes with higher photodegradation yields would also show higher photoreduction rates for TTC.

### 3.3. Possible Reaction Mechanisms

Generally, photocatalytic processes employ semiconductor particles as catalysts. Photoexcitation using light with an energy greater than the band gap promotes an electron from the valence band to the conduction band, creating an electronic vacancy at the valence band edge [70,71,72]. The band edge position of the conduction band and the valence band of the semiconductor, together with the redox potential of the adsorbed substrate, determines whether an oxidative/reductive reaction is thermodynamically allowed. Assuming that the adsorbate possesses an appropriate redox potential, it can be oxidized by transferring an electron to a photogenerated hole, or it can be reduced through the acceptance of an electron. Therefore, the differences in semiconductor particle species and their contents in the ashes lead to different oxidative/reductive abilities.

The possible mechanisms of the featured photoinduced oxidation/reduction may follow the acknowledged “electron–hole pair” theory [73,74]. The presence of tert-butyl alcohol (TBA), a well-known scavenger of hydroxyl radicals, led to some quenching of the photoactivity in most cases (Figure 8). This confirmed the proposition that the degradation of dyes catalyzed by these plant leaf ashes under solar light may partially follow a radical-type mechanism. As is well known, the oxidation of organic dye is mediated by hydroxyl radical, the photooxidation product of H_2_O. However, particular cases, such as couchgrass ash, which showed a high photocatalytic activity even in the presence of TBA, have been observed. It could be inferred that the hydroxyl radical was not the only oxidizing species. For example, organic molecules could be oxidized directly by photogenerated holes.

In the reaction of TTC photoreduction, TTC acts as an electron receptor to estimate DHA, and no other reagents are added into the reaction solution, except plant leaf ashes. This shows that the photogenerated electron induced this reduction reaction. In order to gain a clear idea of the reaction process, the photoreduction of TTC over garlic sprout ashes under analogue solar light irradiation, in the presence of sacrificial reagents, was investigated. As a result, the increase in TTC photoreduction rates when a hydrogen-free sacrificial reagent such as Na_2_SO_3_ was added into the reaction solution (Figure 9) confirmed that the suppression of the electron–hole recombination improved the efficiency of the reduction reaction. Additionally, compared with the results obtained under solar light (Table 3), the photoreduction efficiencies of garlic sprout leaf ash and stem ash were higher, probably caused by the higher light intensity of the Xe lamp than solar light. However, the photoreduction efficiency of garlic root ash decreased. A possible reason is that the ash of the garlic sprout leaf, stem, and root contain different chemicals, meaning that their responses to a change in the light source may not be the same.

Therefore, the diversity of the structure and composition of plant leaf ashes is conducive to visible light catalytic activity. These ashes show photocatalytic activity that may also be explained by the existence of macrochannels that increase the photoabsorption efficiency, and allow the efficient diffusion of dye molecules. Moreover, the different morphologies among the different parts of plants may cause these different performances. Indeed, the mechanism details of these solar-light-driven processes are still far from understood.

Even though the photosynthesis system had been destroyed by calcinations, the inorganic compounds from plant leaves still exhibited the ability to convert solar light. This revealed the existence of another way in which plants use solar energy. At the moment, it is hard to correlate the observations we obtained here directly with photosynthesis, but it did remind us that the photocatalytic activity of inorganic minerals has to be taken into account. We believe that further investigations into the mechanism and other photocatalytic reactions of plant leaf ashes will lead to many interesting results.

Remarkably, the possible contributions of heterogeneous photocatalysis to abiogenesis occurring under the primitive Earth’s conditions [38,39,40], and the photocatalysis and photosorption in the Earth’s atmosphere [75] have been proposed, in addition to its application in environmental remediation and detoxification [69,71,74]. Moreover, UV irradiation may have been the driving and selecting force during the earlier stages of biological evolution, and prebiotic molecules could have been delivered to the early Earth by cosmic particles as the result of interstellar photochemistry [76,77,78]. Photochemical reactions on mineral and metal oxide surfaces may also have contributed to the synthesis of prebiotic molecules on early Earth [79,80,81]. Therefore, the results obtained in this study encouraged us to study their implications for the evolution of photosynthesis. As described above, the oxidizing and reducing powers generated in the presence of minerals and light irradiation, as well as dehydrogenase mimetic activities, suggested that minerals must have catalyzed a series of reactions in the course of chemical evolution and the origin of life.

## 4. Conclusions

Taken together, in the present study, the calcined plant leaf ashes exhibited the solar light photocatalytic activity required to carry out both oxidation and reduction reactions under solar light irradiation. This may encourage us to learn more about the potential roles of minerals in life. The photocatalytic activity of the inorganic minerals in plant leaves after the removal of three key aspects (the antenna, reaction center, and quinine pool) of a primary process in natural photosynthesis implies that there must have been an original photocatalytic process that has maintained evolutionary continuity with contemporary photosynthesis. Particularly, compared to all the chemically synthesized photocatalysts, the plant photocatalysts created via the biosynthesis process are sustainable, and can be produced on a large scale easily. The method would also be green and environmentally benign, by using nontoxic plant leaves as the raw material. Moreover, our findings exemplify how the soluble and insoluble minerals in plant leaf ashes can be synergistically designed to yield efficient, simple, cheap, environmentally friendly, and unique next-generation photocatalysts. They may also lead to advances in artificial photosynthesis and photocatalytic dehydrogenase.

## Figures and Tables

**Figure 1 nanomaterials-13-02260-f001:**
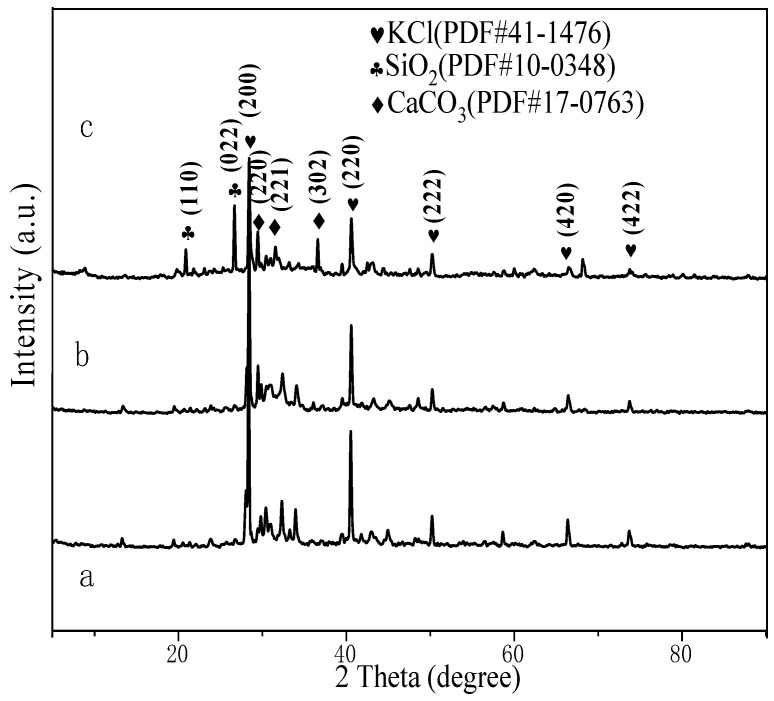
XRD patterns of (**a**) garlic sprout (root) ash; (**b**) garlic sprout (stem) ash; (**c**) garlic sprout (leaf) ash.

**Figure 2 nanomaterials-13-02260-f002:**
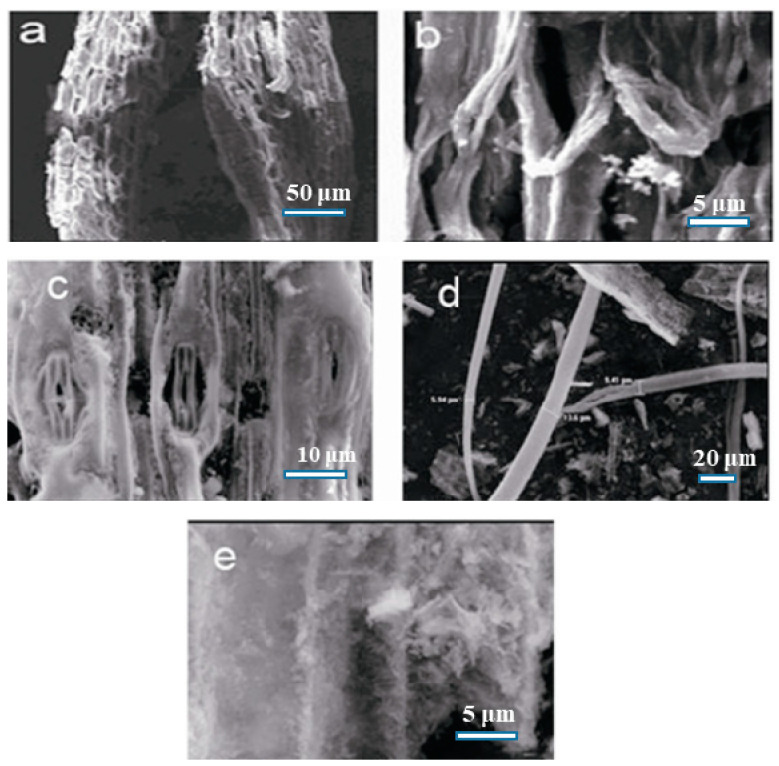
SEM images of ash samples: (**a**,**b**) Garlic sprout (stem); (**c**) couchgrass (*Elymus repens*); (**d**) Bracteata (Hyparrhenia bracteata); (**e**) yellowish leek (stem) (Allium tuberosum Rottler).

**Figure 3 nanomaterials-13-02260-f003:**
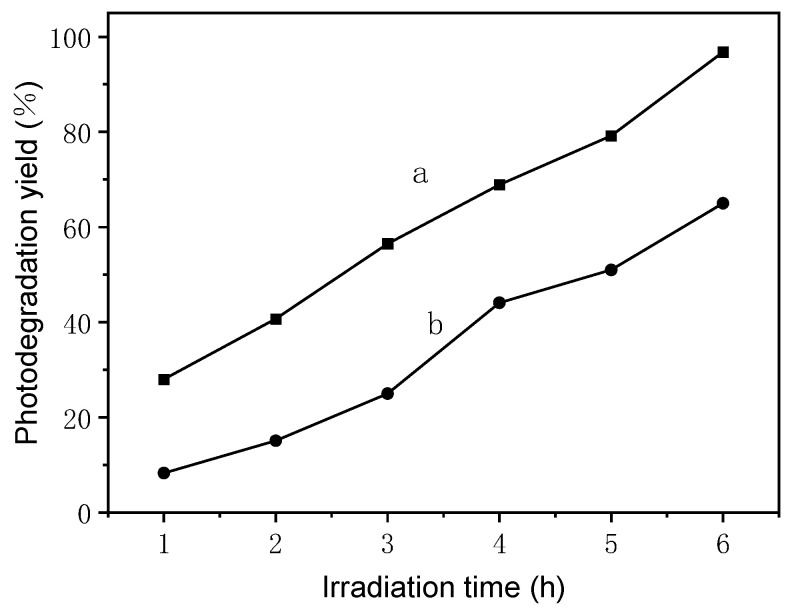
Photodegradation of MB, monitored as the photodegradation yield versus the irradiation time under solar light over the stem ashes of garlic sprouts. (**a**) Garlic sprout stem ashes; (**b**) washed garlic sprout stem ashes.

**Figure 4 nanomaterials-13-02260-f004:**
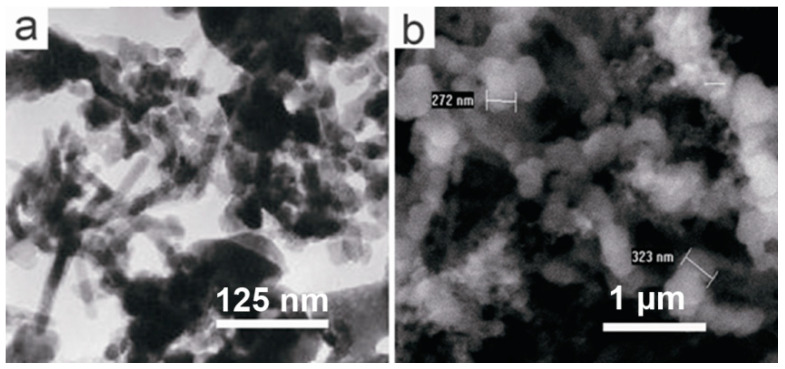
(**a**) TEM image, and (**b**) SEM image of sugarcane (*Saccharum officinarum*) top ash.

**Figure 5 nanomaterials-13-02260-f005:**
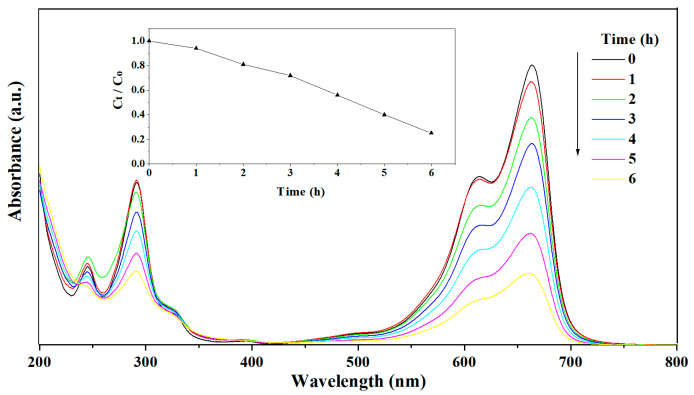
Absorption spectra of a solution of MB in the presence of sugarcane top ash under solar light. Inset shows the concentration change versus irradiation time.

**Figure 6 nanomaterials-13-02260-f006:**
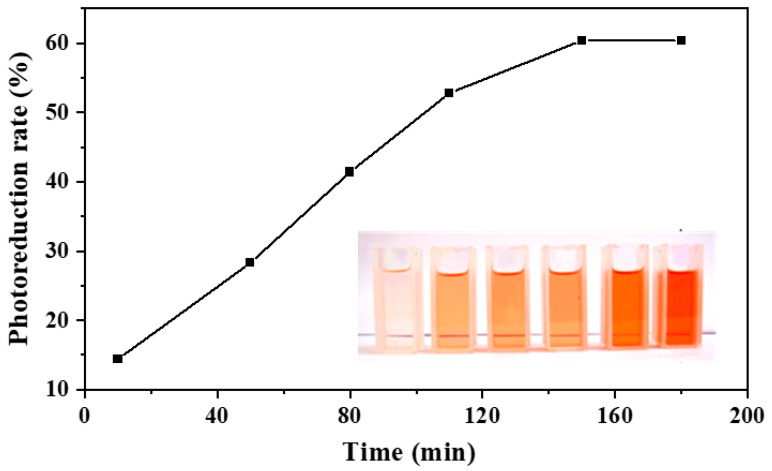
Photoreduction of TTC over bracteata ash under solar light.

**Figure 7 nanomaterials-13-02260-f007:**
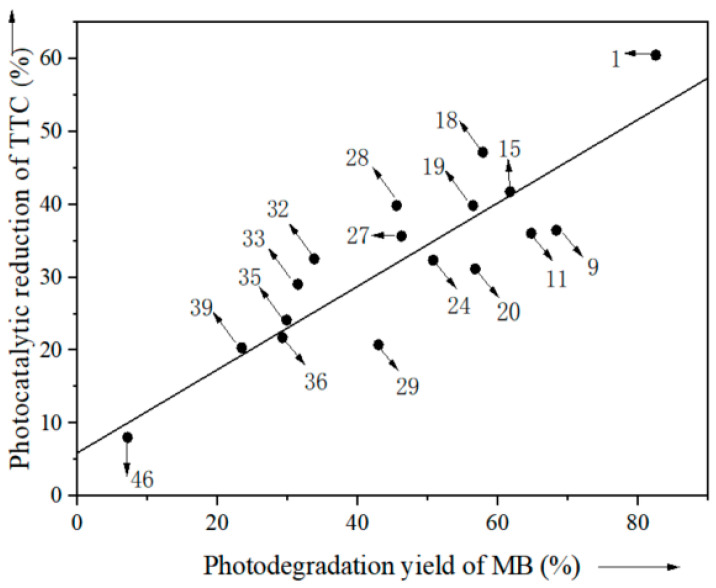
Linear correlation between the photodegradation yield of MB and the reduction of TTC over plant ashes (the numbers in the figure are entries in Table 1).

**Figure 8 nanomaterials-13-02260-f008:**
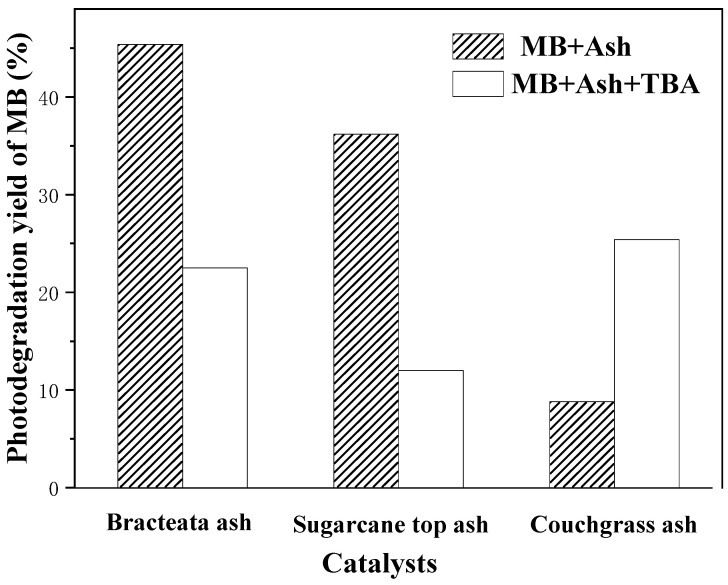
Effects of TAB on the photodegradation yield of MB over plant leaf ashes (reaction conditions: 50 mL 10 ppm MB, 50 mg catalyst, 1.5 mL TAB when used, 260 min irradiation under a 500 W Xenon lamp).

**Figure 9 nanomaterials-13-02260-f009:**
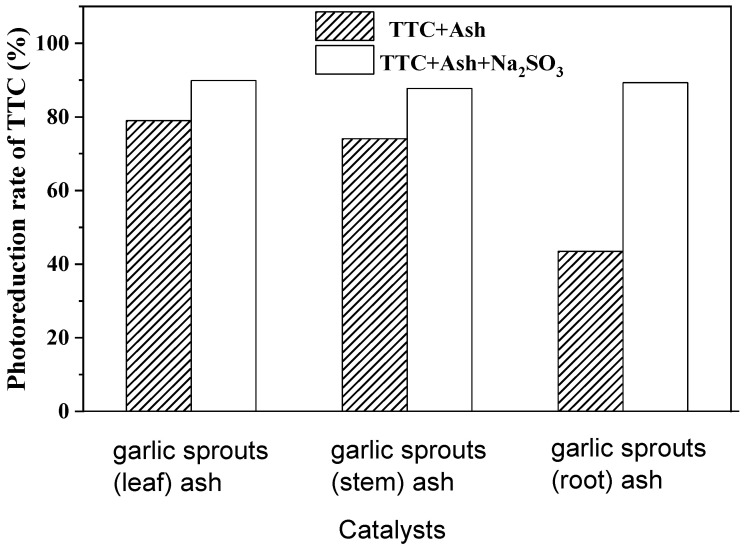
Effects of sacrificial reagents (Na_2_SO_3_) on the photoreduction rate of TTC over garlic sprout ashes (reaction conditions: 50 mL 1 g/L TTC, 25 mg catalyst, 0.1 M Na_2_SO_3_ when used, 3 h irradiation under a 500 W Xenon lamp).

**Table 1 nanomaterials-13-02260-t001:** Summary of the names and Latin names of the plants collected in Yunnan Province.

Entry	Name of Plants	Latin Name of Plants
1	bracteata	Hyparrhenia bracteata (Humb.et Bonpl.) Stapf
2	oyster mushroom	Pleurotus ostreatus fr.
3	couchgrass	Elymus repens
4	edible tree fungus	Auricularia auricula
5	all grass of tigergrass	Thysanolaena maxima
6	sugarcane top	Saccharum officinarum
7	kelp	Thallus Laminariae
8	bamboo	Yunnanensis Hsuch et Yi
9	cycas	Cycas
10	rice hull	Oryza satival
11	giant ambulia	Limnophila aquatica
12	ginkgo biloba L.	ginkgo
13	ivy	Hedera nepalensis
14	ramalina canduplicans	Fructificatio Schizoplylli Communis
15	water lettuce	Pistia stratiotes L.
16	Chinese weeping cypress	Cupressus funebris
17	howard digitaria	Digitaria chinensis hornem
18	crofton weed	Eupatorium adenophorum Spreng
19	furit of cushaw	Cucurbita moschata
20	gastrodia tuber (stem)	Gastrodia elata Blume (stem)
21	burma reed	Neyraudia reynaudiana (Kunth) Keng ex Hitchc
22	azalea (leaf)	Rhododendron (leaf)
23	corn (leaf)	Zea mays Linn
24	pine	pinus
25	giant reed	Arundo donax Linn
26	danhuidajiezhu	Inda sasa Singulispicula
27	gastrodia tuber (flower)	Gastrodia elata Blume (flower)
28	lanceolate arthraxon	Arthraxon lanceolatus (Roxb.) Hochet
29	china loropetal	Loropetalum chinense
30	garlic sprouts (leaf)	Allium sativum (leaf)
31	root of common apluda	Apluaa mutica Linn
32	broad-leaved Indocalamus	Latifo Lius(Keng) Mcclure
33	common duckweed	Lemna minor Linn.
34	reed	Communis Trin
35	long-tongued long bamboo	Dendrocalomus longiligulatus
36	waterweed	Elodea densa
37	herb of common vetch	Vicia sativa L.
38	rice (leaf)	Oryza sativa
39	Japan fatsia	Fatsia japonica
40	garlic sprouts (stem)	Allium sativum (stem)
41	cyanobacteria	Cyanobacteria
42	Chinese leek (leaf)	Allium tuberosum (leaf)
43	azalea (flower)	Rhododendron
44	yeast	Saccharomyces cerevisiae
45	garlic sprouts (root)	Allium sativum (root)
46	bitter bamboo leaf	Pleioblastus amarus (Keng) Keng f. (Arundinaria amara Keng)
47	Chinese leek (stem)	Allium tuberosum (stem)
48	yellowish leek (stem)	Allium tuberosum Rottler (stem)
49	yellowish leek (leaf)	Allium tuberosum Rottler (leaf)

**Table 2 nanomaterials-13-02260-t002:** Main elements content of the garlic ashes, estimated via XRF measurement.

Name of Plants	The Main Element Content (Mass %)							
	K	Cl	O	Ca	C	Mg	Si	B
garlic sprouts (root)	12.57	10.87	38.99	4.89	4.77	4.44	5.08	3.40
garlic sprouts (stem)	27.64	22.70	28.06	6.77	6.24	1.85	0.23	2.11
garlic sprouts (leaf)	26.60	29.20	25.19	5.62	4.74	2.55	0.34	2.28

**Table 3 nanomaterials-13-02260-t003:** Photodegradation yield of MB and photoreduction rate of TTC over plant leaf ashes, under solar light.

Entry	Name of Plants	Photodegradation Yield of MB ^a^ (%)	Photoreduction Rate of TTC ^b^ (%)
1	bracteata	82.6	60.4
2	oyster mushroom	81.8	71.2
3	couchgrass	80.0	25.8
4	sugarcane top	74.4	53.3
5	ramalina canduplicans	62.6	68.3
6	corn (leaf)	51.9	84.5
7	garlic sprout (leaf)	40.7	67.9
8	garlic sprout (stem)	13.1	60.4
9	garlic sprout (root)	7.7	59.9
10	yellowish leek (leaf)	4.4	61.7
11	edible tree fungus	79.3	43.4
12	all grass of tiger grass	76.7	62.4
13	kelp	70.1	36.5
14	bamboo	70	23.4
15	cycas	68.4	36.4
16	rice hull	66.1	7.8
17	giant ambulia	64.8	36
18	ginkgo biloba L.	64.6	25
19	ivy	63.8	26.5
20	water lettuce	61.8	41.4
21	Chinese weeping cypress	61.5	57.3
22	howard digitaria	59.6	52.2
23	crofton weed	57.9	47.1
24	furit of cushaw	56.8	31.1
25	gastrodia tuber (stem)	56.5	39.8
26	burma reed	55.2	52.2
27	azalea (leaf)	54.8	50.7
28	pine	50.8	32.3
29	giant reed	50.7	52.8
30	danhuidajiezhu	48.8	8.8
31	gastrodia tuber (flower)	46.3	35.6
32	lanceolate arthraxon	45.6	39.8
33	china loropetal	43	20.7
34	root of common apluda	36.6	16.1
35	broad-leaved Indocalamus	33.8	32.5
36	common duckweed	31.5	29
37	reed	30.1	34.6
37	reed	30.1	34.6
38	long-tongued long bamboo	29.9	24.1
39	waterweed	29.3	21.7
40	herb of common vetch	24.5	51.6
41	rice (leaf)	24.5	50.7
42	Japan fatsia	23.5	20.3
43	cyanobacteria	12.1	23.1
44	Chinese leek (leaf)	11.9	77
45	azalea (flower)	10.2	33.8
46	yeast	8	7.9
47	bitter bamboo leaf	7.2	7.8
48	Chinese leek (stem)	5.1	67
49	yellowish leek (stem)	-	56.1
	Blank	-	-

Notes: 50 mL 10 ppm MB, 6 h irradiation under solar light. ^a^ 25 mg ash in 50 mL 10 ppm MB, 6 h irradiation under solar light. ^b^ 25 mg ash in 50 mL 1 g/L TTC solution, 3 h irradiation under solar light.

## Data Availability

Not applicable.

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
