# Peer review of "Plant Photocatalysts: Photoinduced Oxidation and Reduction Abilities of Plant Leaf Ashes under Solar Light"

_nanomaterials, 2023, doi:10.3390/nano13152260_

Round 1

Reviewer 2 Report

The paper is devoted to photocatalytic oxidation and reduction activity of leaf ashes obtained by high temperature calcination of the leaves of various plants. Some of studied  ashes showed solar light activity sufficient to degrade methylene blue dye and  reduce the 2,3,5-triphenyltetrazolium chloride. The ideas described in the work are interesting and to a certain extend pioneering.

However, more careful presentation of the results obtained is needed.

- Detailed comments and proposals are available in the attached file of the paper draft.

- The phrase ''Planted photocatalysts'' in the title is intriguing, but a bit inaccurate.

- All statements in the conclusions part have to be commented in the main text.

- Special attention has to be paid on the references list - to be prepared in conformity with the journal requirements.

- For me is unclear why the same figures and tables are given both in the main and supplementary text, except of S1.

Moderate editing of English language required

Reviewer 3 Report

Design of catalysts is one of the important branches of modern science, technology and industry. There is wide spectrum of the ways to design and produce catalysts. Among them, the way offering to use naturally available materials – the plants and their constituents – as an alternative photocatalysts is gaining more attention. Despite of quite extensive research in this field, the issue whether or not the ashes of the plants, when the natural photosynthesis systems is destroyed, are still photocatalytically active is not addressed. The present work fills this gap. Using a large number of calcined samples of various plants as catalysts, and extensively using a complimentary set of characterization tools the authors made novel conclusion relating to photodegradation and photoreduction of MB and TTC. They found that almost all studied samples exhibit quite high activity reaching up to 80%. The morphology of the samples determined by different parts of the plant has an important impact on the photocatalytic activity. As a photodegradation reaction background the authors propose the radical type mechanism, induced by photogenerated careers.

The strength of the work: Quite extensive and comprehensive study based on accurate sample preparation techniques and wide combination of complementary characterization tools, enabling convincing data. The reaction mechanism, albeit qualitative, is proposed. Quite novel character of the data and their high applied relevance for design of environmentally friendly catalysts.

The weakness: The work would gain if there were demonstration of what specific wavelength is responsible for a certain reaction and sample. Also, the XRF measures an average content over a certain volume of the sample. However, as the catalytic activity is determined mainly by the very surface of the sample, it would be informative to find out the elemental and molecular content of the surface, say by ambient-pressure XPS, to refine the catalytic mechanism.

In general, the manuscript is scientifically sound with the appropriate design to address the issues under consideration. It is clear, relevant for the field and presented in a well-structured manner. The manuscript provides sufficient details that support the conclusions. Referencing is quite comprehensive and up-to-dated. The work is suitable for publications in Nanomaterials in its present form.

Round 2

Reviewer 1 Report

The revision is acceptable for publication.